# Amyloids: Regulators of Metal Homeostasis in the Synapse

**DOI:** 10.3390/molecules25061441

**Published:** 2020-03-23

**Authors:** Masahiro Kawahara, Midori Kato-Negishi, Ken-ichiro Tanaka

**Affiliations:** Department of Bio-Analytical Chemistry, Faculty of Pharmacy, Musashino University, 1-1-20 Shinmachi, Nishitokyo, Tokyo 202-8585, Japan; mnegishi@musashino-u.ac.jp (M.K.-N.); k-tana@musashino-u.ac.jp (K.-i.T.)

**Keywords:** calcium, channel, neurotoxicity, zinc, copper

## Abstract

Conformational changes in amyloidogenic proteins, such as β-amyloid protein, prion proteins, and α-synuclein, play a critical role in the pathogenesis of numerous neurodegenerative diseases, including Alzheimer’s disease, prion disease, and Lewy body disease. The disease-associated proteins possess several common characteristics, including the ability to form amyloid oligomers with β-pleated sheet structure, as well as cytotoxicity, although they differ in amino acid sequence. Interestingly, these amyloidogenic proteins all possess the ability to bind trace metals, can regulate metal homeostasis, and are co-localized at the synapse, where metals are abundantly present. In this review, we discuss the physiological roles of these amyloidogenic proteins in metal homeostasis, and we propose hypothetical models of their pathogenetic role in the neurodegenerative process as the loss of normal metal regulatory functions of amyloidogenic proteins. Notably, these amyloidogenic proteins have the capacity to form Ca^2+^-permeable pores in membranes, suggestive of a toxic gain of function. Therefore, we focus on their potential role in the disruption of Ca^2+^ homeostasis in amyloid-associated neurodegenerative diseases.

## 1. Introduction

The oligomerization of proteins affects their conformation, shape, and function. The concept of conformational diseases (or protein misfolding diseases), highlighting the importance of protein conformational changes in the pathogenesis of disease, was first proposed in 1997 [1], and has been supported by recent studies [2,3]. Amyloidosis, a conformational disease, is characterized by the accumulation of amyloids, which are protease-resistant insoluble fibrils with β-sheet structures, in various organs [4]. Amyloidosis is associated with various neurodegenerative disorders, including Alzheimer’s disease (AD), prion disease, and Lewy body disease. In these diseases, while the disease-causing amyloidogenic proteins (β-amyloid protein (AβP) in AD, prion protein (PrP) in prion disease, and α-synuclein in Lewy body disease) differ substantially in primary sequence, they have the common ability to form cytotoxic amyloid fibrils composed of oligomers with β-pleated sheets. Furthermore, other amyloidogenic proteins involved in non-neurodegenerative diseases share common characteristics. In type 2 *diabetes mellitus* (T2DM), the accumulation of the islet amyloid polypeptide (IAPP), also termed amylin, is observed [5]. IAPP is a 37-residue polypeptide and is stored with insulin and Zn^2+^ in the pancreatic β-cell granules. It is widely accepted that the aggregation of IAPP and the resulting β-cell death contribute to the dysfunction in T2DM. Human amylin (hIAPP) forms amyloid fibrils with β-sheet structures, and is toxic to cultured hippocampal neurons and islet cells [6]. In contrast, rat amylin, which exhibits 95% similarity in primary sequence to human amylin, does not form β-sheet structures, and does not cause cytotoxicity. Moreover, IAPP and AβP exhibit similarity in the mechanism of toxicity. Both peptides activate inflammation in microglial cells, and the response induced by AβP was blocked by a nonamyloidogenic IAPP mimic [7]. Metals such as Zn^2+^, Cu^2+^, Fe^2+^, and Al^3+^ binds to IAPP and affect its conformation [8,9].

At least two mechanisms likely underlie the neurodegenerative changes observed in these diseases—the loss of normal protective functions and the gain of toxic functions. In this review, we provide an overview of the molecular mechanisms of neurodegeneration in these amyloid diseases, focusing on the common ability of these amyloidogenic proteins to bind metals and form Ca^2+^-permeable channel-like pores in membranes. Metal is a key determinant of protein conformation and normal brain functions. Interestingly, all of these amyloidogenic proteins are localized at the synapse, where metal ions are abundant, and play critical roles in the regulation of metal homeostasis. Accumulating evidence suggests that the disruption of metal homeostasis is a common feature of neurodegenerative diseases [10,11,12]. Accordingly, we hypothesized that the loss of normal metal regulatory functions may be involved in the pathogenesis of AD, prion diseases, and Lewy body diseases [13]. Here, based on this perspective, we review current studies on the role of these proteins in neurodegenerative diseases. Moreover, we discuss how these amyloidogenic proteins may cause Ca^2+^ dyshomeostasis by forming channel-like pores in membranes. We hypothesize that this gain of toxic function of amyloid oligomers may be a common neurodegenerative mechanism in amyloidsis [14]. Membrane lipid properties, including membrane fluidity and net membrane surface charge, impact the incorporation of peptides into membranes and channel formation [15]. Metals can modulate these processes by affecting the oligomerization of amyloidogenic proteins and by affecting membrane properties. Therefore, we also discuss the importance of lipid–metal, metal–protein, and lipid–protein crosstalk in this review article. These dual neurodegenerative mechanisms highlight the potential contribution of metals in the pathogenesis of AD, prion diseases, and Lewy body diseases. They may also serve as key targets for drug development for these neurodegenerative diseases.

## 2. Neurometals and Amyloidogenic Proteins

### 2.1. Neurometals in the Brain

In the brain, essential trace elements, such as iron (Fe), zinc (Zn) and copper (Cu), play critical roles in various brain functions, including neurotransmitter synthesis, neural transmission, myelination, and protection against reactive oxygen species (ROS). Deficiency of these neurometals is known to impair brain functions and cause learning and memory disorders. The levels and distributions of these neurometals differ in the various brain regions [16]. In whole-brain, Fe is the most abundant neurometal, with a concentration of ~200 μg/g (wet weight). Fe functions as a cofactor in numerous enzymes involved in important cellular processes, including oxidative phosphorylation, mitochondrial energy transfer, the synthesis of neurotransmitters (e.g., dopamine), and myelination [17]. In Fe-deficient children, retardation of mental and physical development is observed [18]. Fe is a redox-active metal, and forms ferric iron (Fe^3+^) and ferrous iron (Fe^2+^). Thus, Fe can generate ROS and can be neurotoxic. Fe in foods is absorbed as Fe^2+^ by binding with divalent metal transporter-1 (DMT1) in the gastrointestinal pathway. Ferroxidases, such as ceruloplasmin, convert Fe^2+^ to Fe^3+^, which is then transported in blood after binding transferrin. Thereafter, Fe^3+^ enters neurons or glial cells via transferrin receptors. Because several enzymes require Fe^2+^ as a cofactor, ferrireductases convert Fe^3+^ to Fe^2+^. Therefore, the ratio of Fe^2+^ to Fe^3+^, as well as total Fe levels, need to be tightly controlled for normal brain functioning.

Zn is the second most abundant element in the brain. Its level is estimated to be ~80 μg/g in total brain. Zn is particularly abundant in the hippocampus, amygdala, cerebral cortex, thalamus, and olfactory cortex [19]. Although much Zn is associated with metalloproteins or metal-related enzymes, a considerable amount of Zn is stored in the presynaptic vesicles of glutamatergic neurons as free Zn ions (Zn^2+^). These neurons secrete Zn^2+^ with glutamate into the synaptic cleft when activated. The secreted synaptic Zn^2+^ binds to various receptors, including N-methyl-d-aspartate (NMDA)-type glutamate receptors, γ-aminobutyric acid (GABA) receptors and glycine receptors. Moreover, accumulating evidence suggests that the synaptic Zn^2+^ plays important roles in neural information processing, synaptic plasticity, and memory formation [20]. Zn deficiency causes dwarfism and immune dysfunction in children, as well as delayed mental and physical development [21,22,23]. Moreover, Zn deficiency in adults produces learning, taste, and odor disorders.

There are two types of Zn^2+^ transporters; ZnT transporters, which facilitate Zn^2+^ efflux when in excess, and Zrt-, Irt-like protein (ZIP) transporters, which facilitate Zn^2+^ influx when deficient [24]. At least 14 different ZnT transporters are present in mammals. Among these, ZnT-1 has a major role in the excretion of excess Zn^2+^, and ZnT-3 regulates Zn^2+^ levels in synaptic vesicles. ZIP transporters control Zn^2+^ influx into cells or subcellular organs such as the Golgi apparatus or endoplasmic reticulum (ER). Disorders of Zn^2+^ transporters have been shown to cause Ehlers–Danlos syndrome and other human diseases [25]. Metallothioneins (MTs), such as MT1, MT2, and MT3, also regulate Zn^2+^ homeostasis [26]. MT3 is of particular interest because it is expressed in the brain, and is decreased in AD patient brains [27,28].

Cu is the third most abundant element in the brain. Cu plays a critical role as a cofactor in key enzymatic reactions in the brain. Cu is a component of ferroxidase (ceruloplasmin) and Cu/Zn superoxide dismutase (Cu/Zn-SOD) [29]. Thus, Cu is essential for the maintenance of Fe homeostasis and protection against ROS. Moreover, some neurons have been reported to contain free Cu^2+^ in synaptic vesicles, as well as Zn^2+^. The synaptic Cu^2+^ is released into synaptic clefts during neuronal firing and binds to NMDA-type glutamate receptors and other receptors. Cu is present as Cu^2+^ and Cu^+^, and therefore, excess free Cu is strictly regulated because it participates in the production of ROS, which can result in cytotoxicity. Both Cu deficiency and overaccumulation caused by a perturbation in Cu transport can lead to severe neurodegenerative disorders, as in Wilson’s disease and Menkes disease [30].

Although the levels of these neurometals in the cerebrospinal fluid (CSF) are reportedly about 1 μM or less [31,32], it is plausible that these metals can be concentrated in synaptic clefts. Because each synaptic cleft is regarded as a small compartment with a 120-nm radius and a 20-nm height, and that the total volume of the synaptic clefts is estimated to be about 1% of the extracellular space of the brain [33]. Indeed, Zn and Cu levels in the synaptic clefts are estimated to be 1–100 μM and ~15 μM, respectively [34,35].

### 2.2. Alzheimer’s Disease and Neurometals

It is widely accepted that the disruption of neurometal homeostasis is involved in the pathogenesis of AD, prion diseases, dementia with Lewy bodies (DLB), and vascular dementia (VD) [10,11,12,36]. Among these, AD, VD, and DLB are classified as senile-type dementia. AD is characterized by the loss of neurons and synapses in the hippocampus or cerebral cortex, the extracellular accumulation of senile plaques, and intraneuronal neurofibrillary tangles (NFTs) [37]. The major constituent of senile plaques is AβP, while that in NFTs is phosphorylated tau protein. Although the cause of AD still remains unclear, the amyloid cascade hypothesis posits that AβP-induced neurodegeneration plays a major role in AD pathogenesis, and is supported by numerous studies [38].

AβP is derived from a large precursor protein (amyloid precursor protein, APP) as a small 39–43-amino acid peptide. The excision of the *N*-terminal of AβP from APP is regulated by β-secretase (BACE), while cleavage of the C-terminal is controlled by γ-secretase. The APP gene is mutated in early-onset cases of AD [39]. Presenilin, a γ-secretase, has been shown to control the production of various truncated AβPs [40]. Mutations in presenilin genes impact the production of truncated AβPs, and are found in cases of early-onset familial AD [41].

AβP(1–40), the first 40 amino acid residues of AβP, was reported in 1991 to cause neurotoxicity in vitro as well as in vivo [42]. AβP tends to form oligomers with AβP-pleated sheet structure in solution. Although the solutions of freshly prepared AβP(1–40) are non-toxic, the solutions of aged AβP(1–40) (aggregated by incubation at 37 °C for several days) cause severe neurodegeneration in cultured rat cortical neurons [43]. The β-sheet content in AβP is correlated with the degree of neurotoxicity [44]. Truncated AβP, such as AβP(1–42), has been shown to seed aggregates and enhance the oligomerization of AβP(1–40) [45].

Recent studies using size-exclusion chromatography, gel electrophoresis and atomic force microscopy have demonstrated that monomers and amyloid fibrils are non-toxic, and that soluble oligomers are synaptotoxic and neurotoxic [46,47]. AβP oligomers derived from the CSF of AD patients induce the loss of dendritic spines and synapses, and the blockade of long term potentiation (LTP). Given that synapses play a key role in memory formation, these observations suggest that synaptic dysfunction induced by AβP oligomers, but not monomers or fibrils, is a critical pathogenetic event in cognitive impairment in AD.

The levels of AβP in the CSF of non-demented elderly subjects and AD patients are similar; however, AβP is also present in the CSF of young individuals [48]. Therefore, factors that influence oligomerization are important for AβP toxicity. These factors include the composition of solvents, temperature, metals, and the oxidation state and racemization of AβP [49,50]. Among the acceleratory factors, metals are known to cause conformational changes in various proteins by binding to amino acids such as His, Tyr, and Arg or with phosphorylated amino acids. Using western blotting and HPLC, we showed that aluminum (Al) significantly enhances AβP oligomerization compared with other metals [51,52]. Zn^2+^ and Cu^2+^ also reportedly bind AβP and accelerate its aggregation [53,54]. Moreover, Mn^2+^ also binds *N*-terminal domain of AβP, enhancing oligomerization [55]. Interestingly, senile plaques are rarely detected in the rodent brain, compared with primate brain [56], despite the fact that primate and rodent AβP proteins differ only in three amino acids (Arg^5^, Tyr^10^, and His^13^). Given that all three of these amino acids are involved in metal binding, it is possible that metals contribute significantly to the oligomerization of AβP and the formation of senile plaques.

APP is also a metal-binding protein, and possesses two Zn^2+^ and/or Cu^2+^ binding domains in its *N*-terminal, and reportedly can convert oxidized Cu^2+^ to the reduced Cu^+^ form [57]. Both Zn^2+^ and Cu^2+^ are implicated in the dimerization and trafficking of APP [58], as well as APP expression, processing, and the production of AβP [59]. Additionally, APP reportedly regulates Fe^2+^ efflux from cells by binding to ferroportin, an iron transporter [60]. Notably, transferrin C2 genotype, iron transporters, and the hemochromatosis gene are important risk factors for AD [61]. The supplementation of Fe has reportedly beneficial roles for cognitive functions in AD patients [62]. Furthermore, Fe regulates the expression of several proteins that possess the iron-response element (IRE) sequence in their mRNAs by binding to iron regulatory proteins (IRPs) [63]. The mRNA for APP contains an IRE domain similar to ferritin and other Fe-binding proteins [64]. Therefore, APP can regulate the levels of these neurometals, and vice versa, these neurometals can control APP expression. Other AD-related proteins, presenilins, are also reportedly metal-binding proteins that are involved in the uptake of Zn^2+^ and Cu^2+^ [60]. Zn^2+^ and Cu^2+^ affect several functions of the presenilins, including trafficking, γ-secretase activity, and APP processing [65].

### 2.3. Prion Diseases and Neurometals

Prion diseases include scrapie in sheep, bovine spongiform encephalopathy (BSE) in cow, Creutzfeldt–Jakob disease (CJD), Gerstmann–Sträussler–Scheinker syndrome (GSS), and Kuru in human. In patients with prion diseases, the spongiform degeneration of glial cells and neurons, and the accumulation of the abnormal scrapie-type isoform of PrP (PrP^Sc^) are commonly observed [66]. Normal prion protein (PrP^C^), a 30–35-kDa cell surface glycoprotein, is widely distributed in the brain. PrP^C^ and PrP^Sc^ share the same chemical characteristics with the same primary sequence, except that PrP^Sc^ is protease-resistant, insoluble, and with a high β-sheet content. The contamination of PrP^Sc^ from foods or iatrogenic factors can promote normal PrP^C^ to misfold and aggregate as PrP^Sc^, resulting in disease. Thus, these diseases are also called transmissible spongiform encephalopathies (TSE), and the conformational change to PrP^Sc^ is crucial for disease pathogenesis.

The link between PrP^C^ and metals is of great interest. Decreased levels of Cu and reduced activity of Cu-dependent enzymes have been reported in the brain of PrP-knockout mice [67]. PrP^C^ attenuates hydrogen peroxide-induced neurotoxicity, as PrP^C^ mimics Cu/Zn-SOD [68]. PrP^C^ reportedly binds four Cu atoms in the octarepeat domain (-PHGGGWGQ-) in its *N*-terminal. Cu can also bind to His^96^ and His^111^ [69]. Huang et al. found that PrP^C^ binds to α-amino-3-hydroxy-5-methylisoxazole-4-propionic acid (AMPA)-type and/or NMDA-type glutamate receptors and affects their function in a Cu-dependent manner [70]. Cu reportedly also affects the expression and cellular trafficking of PrP^C^ [71], as well as physiological functions such as neurite outgrowth [72].

Other metal ions, such as Zn^2+^, Mn^2+^ and Ni^2+^ can bind to PrP^C^ [69]. PrP^C^ is evolutionarily related to ZIP-type Zn^2+^ transporters [73]. PrP^C^ binds to the AMPA-type glutamate receptor in postsynaptic membranes and enhances the cellular uptake of Zn^2+^, and may function as a Zn^2+^ sensor at the synapse [74]. Additionally, PrP^C^ functions as a ferrireductase that converts Fe^3+^ to Fe^2+^ [75], which can be then transported by the DMT1/ZIP14 complex [76]. Indeed, altered Fe metabolism and Fe deficiency are observed in PrP-knockout mice [77], and altered levels of ferroxidase and transferrin in the CSF of CJD patients have also been reported [78]. Intriguingly, several studies suggest that Mn may facilitate prion diseases. Mn enhances the stability of PrP in soils and increases its infectivity [79], and an epidemiological relationship between the pathogenesis of CJD and an imbalance in Mn has been suggested [80].

### 2.4. Lewy Body Diseases and Neurometals

Lewy body diseases include Parkinson’s disease (PD), DLB, and multiple system atrophy. The pathological hallmarks of Lewy body diseases are the abnormal intracellular accumulation of Lewy bodies, which are composed of α-synuclein [81]. α-Synuclein is a 140-amino acid protein that is abundantly localized to presynaptic terminals. α-Synuclein has been shown to play critical roles in synaptic functions and in the maintenance of synaptic plasticity. It was also shown that non-amyloid component (NAC), the α-synuclein fragment peptide, co-accumulates with AβP in senile plaques in AD [82]. Therefore, the oligomerization of α-synuclein is considered to contribute to the pathogenesis of Lewy body diseases.

The link between α-synuclein and metals has been extensively studied, particularly as Fe-rich regions (such as the substantia nigra) are vulnerable in PD, and patients with Mn toxicity exhibits Parkinson-like symptoms. α-Synuclein reportedly binds Cu^2+^ and other metal ions in its N and C-terminal domains [83]. The His^50^ residue plays a key role in the interaction between Cu^2+^ and α-synuclein [84], which is intriguing, given that mutation of this residue is implicated in familial PD. α-Synuclein can also bind to other metals, including Al and Mn, which enhance its oligomerization [85]. Moreover, α-synuclein is a ferrireductase that converts Fe^3+^ to Fe^2+^, similar to PrP^C^ [86]. This indicates that α-synuclein controls dopamine synthesis by providing bioavailable Fe^2+^ to tyrosine hydroxylase and other enzymes. Indeed, the levels of Fe as well as the ratio of Fe^2+^ to Fe^3+^ are reportedly altered in the brains of PD patients [87]. The expression of α-synuclein is controlled by Fe levels, because its mRNA possesses an IRE domain similar to APP and ferritin [88]. Mn reportedly induces the overexpression of α-synuclein [89]. 

### 2.5. Hypothesis: Loss of Normal Regulatory Functions of Amyloidogenic Proteins in the Synapse

As described in the previous section, amyloidogenic proteins share several common characteristics. All of these amyloidogenic proteins or related proteins (APP or presenilins) possess metal-binding domains and play major roles in the maintenance of metal ion homeostasis. Furthermore, all of these proteins are reportedly co-localized at the synapse, where metals are abundantly present, and which is a major site of degenerative change in these diseases. Although some APP reportedly is present in the postsynaptic membrane and regulates spine formation [90], it is primarily localized to the presynaptic membrane [91]. PrP^C^ is localized to the postsynaptic membrane with receptors [92], and α-synuclein is mainly found in the presynaptic cytosol. Presenilins are mainly localized in the ER, and in the presynaptic and postsynaptic membranes.

We have developed a hypothetical model of the crosstalk between these amyloidogenic proteins and metals (Figure 1). During neuronal excitation, Zn^2+^ and/or Cu^2+^ are released into the synaptic clefts. The level of synaptic Zn^2+^ is mainly controlled by ZnT-1 transporter, which is primarily localized at postsynaptic membranes [93]. PrP^C^ acts as an analogue of ZIP transporters and also contributes to the regulation of synaptic Zn^2+^ levels. APP and PrP^C^ regulate the levels of Fe^2+^ and Cu^2+^ in the pre-synapse and post-synapse, respectively. Presenilins also participates in the maintenance of Cu^2+^ and Zn^2+^. α-Synuclein controls the Cu^2+^ or Mn^2+^ levels in the presynaptic domains. PrP^C^ functions as a ferrireductase to convert Fe^3+^ to Fe^2+^ in the postsynaptic domain, and regulates Fe^2+^ influx through the DMT1/ZIP14 complex [76]. α-Synuclein acts as a ferrireductase presynaptically. Both proteins control neurotransmitter synthesis and other Fe^2+^-requiring functions in the pre-synapse and post-synapse. In turn, the expression of these amyloidogenic proteins is controlled by the levels of these neurometals.

Given the narrow distance between the pre- and postsynaptic membranes (~20 nm), it is plausible that these proteins can interact with each other. Indeed, PrP^C^ reportedly binds to AβP oligomers and controls its toxicity [94]. α-Synuclein influences the processing of APP as well as AβP secretion [95]. Meanwhile, AβP promotes seeding and spreading α-synuclein [96]. It is also possible that PrP^C^ affects the production of AβP by providing Cu to APP. Carnosine (β-alanyl histidine), which is synthesized and secreted from glial cells [97,98], and MT3 [27,28] may also participate in the maintenance of metal homeostasis at the synapse.

The crosstalk between metals and amyloidogenic proteins described here is highly complex. Therefore, when crosstalk is perturbed by genetic or environmental factors, the resulting disruption of metal homeostasis can trigger various adverse changes and cause neurodegenerative diseases. For example, pathogenetic PrP^Sc^ from contaminated foods causes the depletion of neuroprotective PrP^C^, and initiates oxidative damage by increasing Cu and Fe, thereby increasing susceptibility to ROS, causing the depletion of neurotransmitters, and inducing the degeneration of neurons and glial cells. The metal imbalance caused by excess Mn^2+^ triggers the toxic accumulation of Fe by affecting IRE–IRP binding and by inhibiting the expression of APP and ferritin [99]. 

APP is mainly localized to the presynaptic membrane. PrP^C^ is located in the postsynaptic membrane bound to receptors. APP binds to Cu^2+^ and/or Zn^2+^ and has the ability to convert Cu^2+^ to Cu^+^. APP also regulates Fe^2+^ efflux from cells via ferroportin. Presenilins function as γ-secretases, and affect the uptake of Zn^2+^ and Cu^2+^. PrP^C^ binds to Cu, Zn, and Fe and regulates their levels at the synapse. Additionally, PrP^C^ acts as a ZIP Zn^2+^ transporter analogue, and ZnT-1 Zn^2+^ transporter is also localized to postsynaptic membranes; both control Zn^2+^ levels at the synapse. α-Synuclein is mainly localized to the presynaptic domain, and binds Cu, Mn, and Fe. Both PrP^C^ and α-synuclein have ferrireductase activity, and provide bioavailable Fe^2+^ to enzymes at the pre- and post-synapse, respectively. Fe^2+^ is transported into cells by the complex of ZIP-14 and DMT-1. Other metal-binding factors such as MT3 and carnosine (Car) are secreted into synaptic clefts and play critical roles in the maintenance of metal homeostasis.

NMDA-R; NMDA-type glutamate receptor, AMPA-R; AMPA-type glutamate receptor, FPN: ferroportin; PS: presenilins; colored circles represent Zn, Cu, Fe, Mn, Ca, and glutamate.

## 3. Neurotoxicity by Amyloidogenic Protein Oligomers

### 3.1. AβP-Induced Ca Dyshomeostasis

The preceding discussion focused on neurodegeneration caused by the loss of the normal function of amyloidogenic proteins (Figure 1). Now, we focus on neurodegeneration caused by the gain of neurotoxic function of the amyloid oligomers of these proteins. Oligomerized AβP reportedly has neurotoxic effects, including synaptotoxicity [100]. PrP^Sc^ as well as its peptide fragment (PrP106–126) cause synaptotoxicity and cellular toxicity in vitro and in vivo [101]. Oligomerized α-synuclein is also toxic, and its fragment peptide (NAC) causes neurotoxicity [102].

The molecular mechanism of neurodegeneration induced by oligomerized proteins is of considerable interest. Numerous studies have demonstrated that AβP induces various adverse changes, such as ROS production, cytokine induction, induction of ER stress, and disruption of Ca^2+^ homeostasis [103,104]. Among these, the disruption of Ca^2+^ homeostasis, especially the abnormal increase in intracellular Ca^2+^ levels ([Ca^2+^]_i_) is considered a key pathogenetic event, because Ca^2+^ ions play critical roles in normal brain functions. Perturbed Ca^2+^ homeostasis leads to various adverse events, including the activation of the calpain and caspase pathways, and disruption of numerous key enzymatic reactions. Ca^2+^ is also involved in the processing of APP, as well as tau phosphorylation [105]. The Ca^2+^ dyshomeostasis ultimately leads to membrane disruption, lipid peroxidation, ROS production, and apoptosis. These changes are also observed after exposure to AβP. The disruption of Ca^2+^ homeostasis and the elevated [Ca^2+^]_i_ have been reported in AD patients and in AβP-exposed neurons [106,107,108,109].

Studies by others and our group suggest that AβPs directly incorporate into the membrane and form channels (pores) that allow the passage of Ca^2+^ and other cations, and that the resulting change in [Ca^2+^]_i_ may be a primary pathogenetic event in AβP neurotoxicity [110,111,112]. Neurotoxic peptide fragments of AβP, including AβP(1–40), AβP(25–35) and AβP(1–42), have been demonstrated to incorporate into artificial planar lipid membranes and form giant multi-level pores (~5 nS) with cation (including Ca^2+^) selectivity [113,114,115]. To assess the validity of this amyloid channel hypothesis, we performed an electrophysiological study to examine whether AβPs form channels on neuronal cell membranes as well as on liposomes. For this experiment, we employed membrane patches from immortalized hypothalamic neurons (GT1–7 cells), which are derived from murine hypothalamic neurons [116]. Within 3–30 min of the addition of AβP(1–40), a channel current was detected across the excised membranes as well as the liposome membranes. The activity of the channel in both preparations was inhibited by the addition of Zn^2+^, and recovered by a Zn^2+^ chelator, *o*-phenanthroline [117,118]. AβP(1–40), as 5–8-mer oligomers, forms pore-like structures in the membranes, as revealed by 3-D computer simulations [119]. Nuclear magnetic resonance (NMR) studies show that AβP(1–42) pores are composed of tetrameric and hexameric oligomers [120]. Voelker et al. demonstrated that AβP(1–40) as well as AβP(1–42) formed trimers to pentamers that could form ion channels in water [121]. AβPs were shown to form pores by high resolution atomic force microscopy (AFM) as well [122]. Lee et al. observed the formation of AβP channels on membranes from brain total lipid extracts using AFM [123]. Pores or pore-like structures consisting of AβP have also been observed in AD patients [124,125,126].

We also demonstrated that exposure to AβP causes an increase in [Ca^2+^]_i_ in GT1–7 cells using multi-site fluorometry with fura-2, a Ca^2+^-sensitive fluorescent dye [15,127,128,129,130]. A robust increase in [Ca^2+^]_i_ occurs in many GT1–7 cells after exposure to AβP(1–40) (Figure 2A). Figure 2B shows the temporal changes in [Ca^2+^]_i_ in these cells during the exposure. Exposure to both AβP(1–40) (*line (a)*) and AβP(1–42) (*line (c)*) causes a marked increase in [Ca^2+^]_i_. However, AβP(40–1), a reverse-sequence peptide with no neurotoxicity, caused no change in [Ca^2+^]_i_ (*line (b)*).

Despite the empirical evidence, the amyloid channel hypothesis still remains controversial. There are several other mechanisms by which AβPs could interact with neurons and disrupt calcium homeostasis. These include the activation of some type of cell surface receptor coupled to Ca^2+^ influx, and the disruption of membrane integrity [107]. AβPs were reported to bind to NMDA and AMPA glutamate receptors [130], as well as nicotinic acetylcholine receptors [131], and all of these receptors are highly Ca^2+^-permeable. Furthermore, AβP reportedly influences voltage-gated Ca^2+^ channels and the inositol triphosphate (IP_3_) receptor [132]. Furthermore, presenilins have been implicated in capacitative Ca^2+^ entry, ER Ca^2+^ signaling, and mitochondrial Ca^2+^ signaling, and that mutations in these proteins affect Ca^2+^-regulated functions [133].

Nonetheless, despite the controversy, we have demonstrated that D-AβP(1–40), composed of D-amino acid residues only, also caused an increase in [Ca^2+^]_i_, similar to L-AβP(1–40) (*line (d)*). D-AβP(1–40) reportedly possess similar toxicity to L-AβP(1–40) [134], as well as the ability to form pores [135]. The AβP-induced increase in [Ca^2+^]_i_ was not influenced by the addition of the Na^+^ channel blocker tetrodotoxin, the Ca^2+^ channel blocker nifedipine, an antagonist of the NMDA receptor (D-APV), or an antagonist of the GABA receptor (bicuculline) [15]. Moreover, our detailed quantitative analysis using multi-site Ca^2+^ fluorometry suggests the AβP-induced rise in [Ca^2+^]_i_ was highly heterogeneous among the GT1-7 cells, although this cell line is genetically homogenous. As shown in Figure 2A, the exposure to the same peptide solution produced different changes in the magnitude and latency in [Ca^2+^]_i_ in cells in the same field of view [15,129]. This suggests that AβP-induced [Ca^2+^]_i_ changes are mediated by unregulated amyloid channels, and not by endogenous receptor-mediated pathways.

Membrane properties, such as the surface net charge and fluidity, influence the interactions between proteins and membranes, and accordingly, channel formation [136]. In particular, the negative charge of the membrane outer surface is critical for the incorporation of AβP, as the peptide normally possesses a positive charge [137]. The ratio of cholesterol to phospholipids alters the membrane fluidity and influences the oligomerization process as well. We have demonstrated that the pre-administration of several compounds that affect membrane fluidity and membrane potential, including cholesterol, phloretin, and estradiol, markedly inhibit AβP-induced [Ca^2+^]_i_ increases in GT1-7 cells [128,129]. Gangliosides, which possess a negative charge (from their sialic acid residues), may influence these processes. GM1 gangliosides were reported to bind to AβP in AD brains [138]. It was also suggested that GM1-bound AβP behaves as a “seed” for oligomers and enhances AβP oligomerization [139,140]. Our AFM imaging study demonstrated the deposition of AβP(1–40) on ganglioside (GM1)/phospholipid monolayers [15].

### 3.2. Channel Formation by Other Amyloidogenic Proteins

The ability to form Ca^2+^-permeable pores is also found in certain toxins and venoms. The α-toxin of Staphylococcus aureus possesses hemolytic activity by forming pore-like hexamers composed of 33-kDa subunits with β-sheet structures, which allow Ca^2+^ influx [141]. Other antimicrobial peptides, such as magainin 2 (a 26-residue antimicrobial peptide obtained from the skin of *Xenopus laevis*), melittin (a 28-residue peptide from bee venom), and antibiotics such as amphotericin and gramicidin, were also reported to form Ca^2+^-permeable pores on cell membranes and to cause cell lysis [142]. The electrophysiological activity of gramicidin and amphotericin on membranes excised from hippocampal neurons was shown to be similar to that of AβP(1–40) [143]. Thus, it is possible that AβP has similarities to these pore-forming venoms, toxins, and antibiotics. Indeed, AβP reportedly possesses antimicrobial activity [144].

Accumulating evidence demonstrates that other amyloidogenic proteins, including PrP^Sc^ and α-synuclein, form amyloid channels (pores). Similar to AβP, PrP106–126 reportedly forms cation-permeable pores in artificial lipid bilayers [145]. Kourie et al. demonstrated that channels formed by PrP106–126 are Cu-sensitive cation channels [146]. Lashule et al. observed that α-synuclein forms annular pore-like structures similar to AβP(1–40) by electron microscopy [147]. The pore formation by α-synuclein on mitochondrial membranes was reportedly accelerated by cardiolipin, a membrane lipid [148].

Other amyloidogenic proteins involved in non-neurodegenerative diseases can form channel structures as well. Human amylin (hIAPP) reportedly forms ion channels on liposomes, meanwhile non- toxic rat amylin does not form ion channels [149]. Similar to AβP, the membrane composition such as cholesterol and raft affects fiber growth of IAPP and the membrane damage [150,151]. It was also demonstrated that the membrane curvature plays critical roles in the amyloid formation of IAPP and the resulting membrane damages [152]. Calcitonin, linked with medullary thyroid carcinoma, also forms pore-like structures on membranes [153]. The AFM imaging study demonstrated that α-synuclein, hIAPP, and other amyloidogenic proteins form annular pore-like structures on bilayer membranes, similar to AβP [154]. We found that these amyloidogenic proteins also cause an increase in [Ca^2+^]_i_, similar to AβP(1–40), as shown in Figure 2B. A marked increase in [Ca^2+^]_i_ is observed after exposure to PrP106–126 (*line (e)*), hIAPP (*line (g)*), NAC(*line (i)*), and magainin 2 (line (j)). However, control peptides such as scrambled PrP106–126 (a peptide with a random sequence; *line (f)*) and rat amylin (*line (h)*) did not cause notable changes. Furthermore, PrP106–126, hIAPP and AβP(1–40) induce membrane disruption in liposomes [15]. 

### 3.3. Hypothesis: The Gain of Toxic Functions of Amyloid Oligomers at the Synapse

Based on the findings described above, we propose the amyloid channel hypothesis for AD pathogenesis (schematically shown in Figure 3). We also posit that similar mechanisms may underlie other diseases, such as prion diseases and Lewy body diseases, given their commonalities.

AβPs are secreted from APP into synaptic clefts, and are usually degraded rapidly by proteases. However, increased AβP secretion or an altered ratio of AβP(1–42) to AβP(1–40), caused by mutations in APP and/or presenilins, may increase the half-life (and accordingly, the concentration) of AβP in the brain. The contamination of toxic Al causes the up-regulation of APP and the increased amount of AβP [155]. The accumulated AβP peptides then aggregate and form pore-like channels on membranes. Because these AβP channels are unregulated, Ca^2+^ flows freely through them, eventually triggering various neurodegenerative processes including ROS formation, disturbances in capacitive Ca^2+^ in the ER, and tau phosphorylation. Additionally, the elevation in [Ca^2+^]_i_ can result in the overproduction of APP and the increased production of AβP, ultimately inducing a vicious cycle of unregulated Ca^2+^ influx.

Our hypothesis can explain the long delay in AD development. Although AβP is normally secreted in young and in non-demented elderly subjects, AD occurs only in senile subjects. Net charges on the membrane outer surface influence the rate of channel formation by affecting the binding of proteins. Given that positively charged and neutral phospholipids (e.g., phosphatidylcholine) are normally present on the outer membrane surface, the membrane in normal and young brains seldom bind AβP, because it possesses a positive charge. However, when the asymmetric distribution is disrupted during apoptotic death, and the negatively charged phospholipids, such as phosphatidylserine, are exposed on the surface, AβP peptides can easily bind membranes and form channels. This hypothesis may also explain the contribution of various environmental factors as well as genetic factors to AD pathogenesis. It is well known that the ratio of cholesterol to phospholipids is altered by various environmental and genetic factors (e.g., foods, apolipoprotein E genotype), and is linked with AD pathogenesis. Cholesterol alters membrane fluidity and influences protein oligomerization, thereby impacting channel formation. Polyunsaturated fatty acids such as docosahexaenoic acid (DHA) and eicosapentaenoic acid (EPA) also affect these processes by altering membrane fluidity. We demonstrated that dehydroepiandrosterone (DHEA), an endogenous testosterone, attenuates the AβP-induced increase in [Ca^2+^]_i_ by influencing membrane fluidity [129].

Because AβP dimers form toxic protofibrils rapidly, compared with monomeric AβP [156], it is possible that metals such as Al^3+^, Zn^2+^, Cu^2+^ and Mn^2+^ accelerate oligomerization and thereby increase the rate of channel formation. The involvement of metals in pore formation by AβP and neurotoxicity is complex. Recent studies demonstrate that not all oligomers are equally neurotoxic, although all possess AβP-pleated sheet structure. The characteristics (size and shape) of AβP oligomers formed in the presence of Al^3+^, Zn^2+^, Cu^2+^ and Fe^3+^ are unique by morphological analysis using AFM [157]. Moreover, Sharma et al. revealed that Zn^2+^-aggregated AβPs are less toxic than Cu^2+^-aggregated AβP [158]. Although Cu^2+^ accelerates the oligomerization of AβP, it reportedly inhibits the oligomerization of human amylin and PrP106-126 [9,159]. In addition, our group and others have demonstrated that the amyloid channel is blocked by Zn^2+^ [116,117]. Given that Zn^2+^ is secreted into synaptic clefts in an activity-dependent manner, Zn^2+^ may therefore protect neurons against AβP toxicity [160]. Similar biphasic effects of Zn^2+^ is observed also in T2DM. Zn^2+^, which is co-secreted with insulin and IAPP from the pancreatic β cells, has a dual effect on the aggregation and cell toxicity of IAPP in a concentration-dependent manner [161]. Zn^2+^ at the physiological concentration inhibits the aggregation of IAPP, meanwhile, atypical (low or high) concentrations of Zn^2+^ enhance the aggregation. 

Metals modulate the interaction between lipids and amyloidogenic proteins. Because gangliosides are negatively charged, the lipid raft may provide a platform for oligomerization. Indeed, Zn^2+^ or Cu^2+^ reportedly influences membrane fluidity [162,163]. Curtain et al. showed that Zn^2+^ and Cu^2+^ affects the membrane insertion of AβP peptides [164]. Membrane-bound metal ions such as Ca^2+^, Mn^2+^ and Zn^2+^ are essential for the toxicity of pore-forming toxins [165].

In pathological conditions, such as the genetic overexpression of APP accumulated AβP peptides can directly incorporate into membranes and form amyloid channels. Metals such as Al^2+^, Zn^2+^, Cu^2+^ and Mn^2+^ enhance the oligomerization of AβP and facilitate the process. Thereafter, disruption of Ca^2+^ homeostasis by these amyloid channels triggers various apoptotic pathways, and causes synaptotoxicity and neurotoxicity. These changes contribute to the pathogenesis of AD. Colored circles represent Zn, Cu, Ca, Mn, and Al.

## 4. Conclusions

Our working hypothesis, shown in Figure 1 and Figure 3, may explain the two aspects of neurodegeneration—loss of the normal metal regulatory function of amyloidogenic proteins, and gain of the toxic function of amyloidogenic proteins. As discussed here, the crosstalk between neurometals and amyloidogenic proteins at the synapse is highly complex and vulnerable to perturbation. The loss of normal functions of amyloidogenic proteins causes the disruption of metal homeostasis, resulting in neurodegeneration. Furthermore, the conformational change in amyloidogenic proteins and the formation of unregulated Ca^2+^ channels is a gain of toxic function that contributes to the pathogenesis of these neurodegenerative diseases. These dual facets of neurotoxicity highlight the major role of trace elements in the pathogenesis of amyloid-associated neurodegenerative diseases. Recently, La Rosa et al. exhibited the similarities of oligomerization patterns and structures among hIAPP, AβP and other amyloidogenic proteins using the theoretical model [166]. These similarities in the conformations of various amyloidogenic proteins may strengthen our hypothesis about the common toxic mechanism in amyloidosis. 

Our hypothesis also has important implications for the development of drugs for neurodegenerative diseases. It is possible that compounds that regulate metal homeostasis and block amyloid channels may have potential for the treatment of neurodegenerative diseases. Indeed, we found that carnosine attenuates Zn^2+^-induced neurotoxicity [167]. Given that carnosine inhibits oligomerization and attenuates AβP-induced neurotoxicity [168] and PrP106–126-induced neurotoxicity [159], it may have therapeutic potential for amyloid diseases. Further research is needed to test this working hypothesis, particularly about the link between lipids and metals.

## Figures and Tables

**Figure 1 molecules-25-01441-f001:**
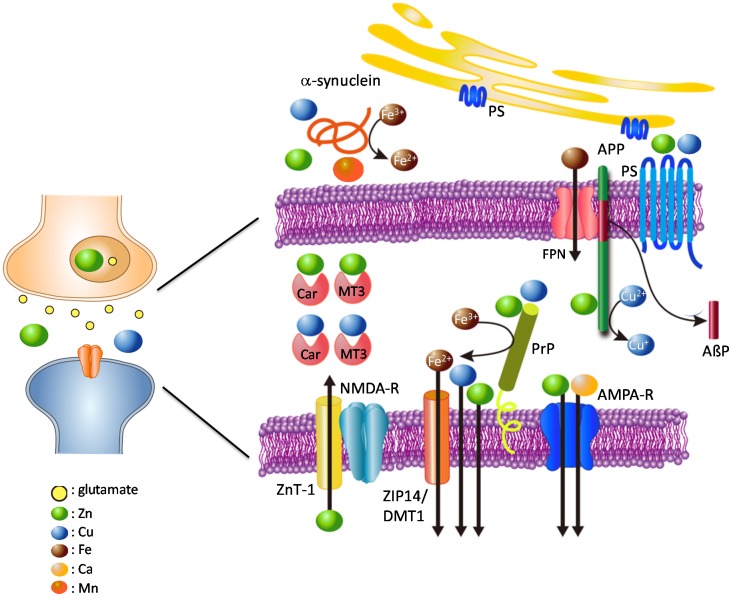
Hypothesis: loss of normal functions of amyloidogenic proteins at the synapse.

**Figure 2 molecules-25-01441-f002:**
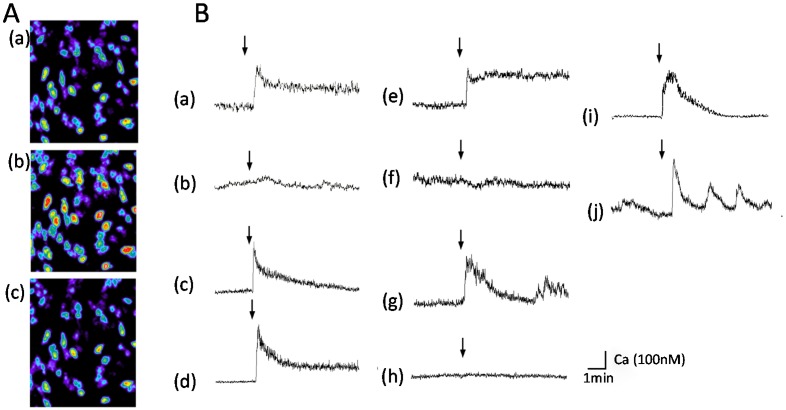
Temporal changes [Ca^2+^]_i_ in GT1-7 cells after exposure to amyloidogenic proteins. (**A**) Temporal changes in [Ca^2+^]_i_ shown as pseudocolor images. AβP(1–40) (10 µM) was applied to GT1–7 cells, and temporal changes in fluorescence intensities corresponding to changes in [Ca^2+^]_i_ were analyzed by multi-site fluorometry using fura-2. (a): 1 min before exposure to AβP(1–40); (b): 20 s after exposure; (c): 5 min after exposure. (**B**) Typical time course of [Ca^2+^]_i_ changes before and after exposure to each amyloidogenic protein (10 µM). (a) AβP(1–40); (b) AβP(40–1); (c) AβP(1–42); (d) D-AβP(1–40); (e) PrP106–126; (f) scramble PrP106–126; (g) human amylin (hIAPP); (h) rat amylin; (i) NAC; (j) magainin 2. The arrow indicates the time of peptide addition (modified from Ref. No. 15 with permission).

**Figure 3 molecules-25-01441-f003:**
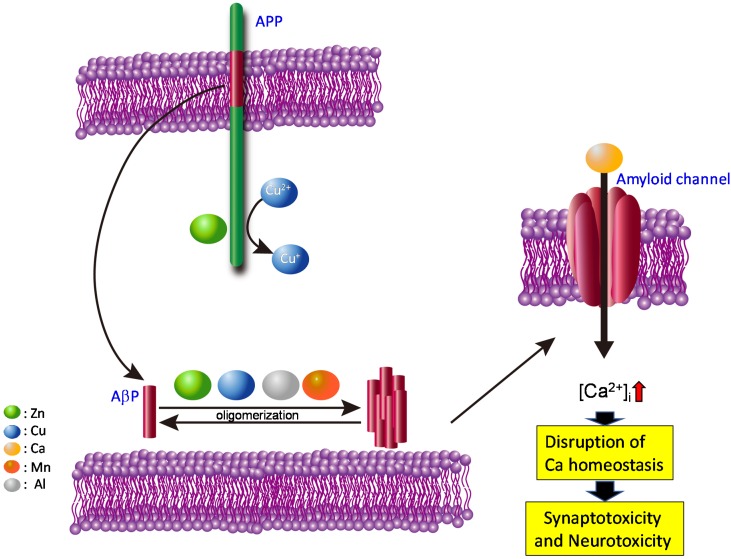
Hypothesis: gain of toxic functions of amyloid oligomers at the synapse.

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
