# Peer review of "Amyloids: Regulators of Metal Homeostasis in the Synapse"

_molecules, 2020, doi:10.3390/molecules25061441_

Round 1

Reviewer 1 Report

The current manuscript entitled “Amyloids: regulators of metal homeostasis in the synapse” aims to review the roles of proteins implicated in neurodegenerative diseases, that share common characteristics including metal binding activity, amyoidogenicity, cytotoxicity and pore formation. The authors propose schemes by which loss of their normal regulatory functions my contribute to neurodegeneration, and how gain of function in the oligomeric form may disrupt calcium homeostasis due to formation of Calcium permeable pores at the plasma membrane.

This manuscript needs proof-reading and editing for correct English. For example, the very first sentence (and line 34) states that the primary sequences of disease-related proteins of several neurodegenerative diseases (such as beta amyloid protein, prion protein and alpha synuclein) are identical. This statement is clearly false, as illustrated in their table 1, however based on the context it is likely a mis-translation. Perhaps they mean "unique"?

On further investigation, it appears that the same authors have published a very similar review in 2017 in the journal Metalloproteins entitled “Cross talk between neurometals and amyloidogenic proteins at the synapse and the pathogenesis of neurodegenerative diseases”. The current manuscript and ideas within overlaps extensively with the previous review (the previous one being comparatively better written) including figures and tables. For example Table 1 appears to be a direct copy/paste from the previous article, excluding a column relating to calcium.

My suggestion would be for the authors to have the manuscript proofread/edited, and for them to provide a summary of how their new review builds on their previous review.

Author Response

Thank you very much for the comments and suggestions. Considering your comments, I re-organized and polished up my manuscript, and I believe, it can be improved.

>This manuscript needs proof-reading and editing for correct English.

    Thank you very much for your suggestion. I asked to correct my english by Edanz (www.edanzediting.com/ac, shown in Acknowledgement).

> for them to provide a summary of how their new review builds on their previous review Thank you very much. I apologize to you for lack of clarity in my manuscript. In the previous review, we had focused the loss of normal protective functions as common neurodegenerative mechanism. In this manuscript, we updated the loss of function hypothesis based on recent literatures, and combined the gain of toxic function hypothesis based on the common features of Ca-permeable pores. I re-organize the most part of Introduction (line 41-45, line50-64). I also corrected Abstract and added the newer explanations about lipid-metal interactions and about metal contribution in the hypothesis (line439-455). To avoid the similarity, I deleted Table1.

Reviewer 2 Report

The review of Tanak et al summarizes the link of the metal homeostasis and amyloidogenic proteins in the current literature. The review has a clear focus and is mainly comprehensive and up to date. It is clearly written; however, I suggest that a native speaker should go through the manuscript. The review is interesting for a broad readership dealing with metal-protein interactions and neurodegeneration in general. It has to be mentioned that several similar reviews already exist dealing with Alzheimer’s disease and ions, nevertheless, the review also includes Lewy Body disease, prion disease and not only Alzheimer’s disease and nicely points out similar mechanisms of action in respect of metal homeostasis. In my opinion, the review is therefore in the focus of the journal and sufficiently comprehensive and new.

I have some minor suggestions before the review is suitable for publication in my opinion.

Whereas the authors clearly point out that manganese is involved in prion and Lewy body disease this aspect is completely missing in respect to Alzheimer´s disease. Newer studies by NMR spectrometry revealed that also manganese ions are able to bind to the N-terminal part of Abeta 1-40 (e.g. J Trace Elem Med Biol, 38, 183-193

Characterization of Mn(II) Ion Binding to the Amyloid-β Peptide in Alzheimer's Disease)

The pore formation ability of Abeta is controversially discussed. Please discuss this aspect more balanced e.g. by citing literature or results not agreeing to this hypothesis. The authors state in some paragraphs that Abeta fibrils are crucial for Alzheimer’s disease pathogenesis (e.g. Abstract lien 12). The current literature emphasizes the fact that more oligomers contribute to the pathogenesis of Alzheimer’s disease. For example, several studies show that the plaque burden only weak or does not correlate with the cognitive decline in Alzheimer´s disease. Please clarify this aspect. I would appreciate if the authors could include or discuss the effect of metal ions on the lipid-Abeta interaction which might be also important for aggregation or the progression of the disease (e.g. Journal of Biological Chemistry 278(5):2977-82)

Author Response

Thank you very much for your comments and suggestions. Considering your comments, I re-organized and polished up my manuscript, and I believe, it can be improved.

>this aspect is completely missing in respect to Alzheimer´s disease. Newer studies by NMR spectrometry revealed that also manganese ions are able to bind to the N-terminal part of Abeta 1-40

     Thank you very much for your suggestions. I am sorry for the overlook about Mn. I added this reference and other references about Mn (Refs. 50, 74,75, 93), and comments about the implication of Mn in our hypothesis (line164-165, 210-212, 268-269). I also corrected Fig 3 and added Mn in the figure. Thanks to your comments, I believe, our hypothesis became much better.

>The pore formation ability of Abeta is controversially discussed. Please discuss this aspect more balanced e.g. by citing literature or results not agreeing to this hypothesis.

     Thank you very much for your comments. I added the controversial explanations about amyloid channel hypothesis such as binding of Abeta to some receptors, in line 332-340 and added Refs. 101, 124-127. I also discussed about the controversy more precisely (line 346-352).

> The current literature emphasizes the fact that more oligomers contribute to the pathogenesis of Alzheimer’s disease. For example, several studies show that the plaque burden only weak or does not correlate with the cognitive decline in Alzheimer´s disease. Please clarify this aspect.

     I am sorry for the lack of clarity in this point. As the reviewer suggested, too much aggregated Abeta is not toxic. I added the precise explanations in line149-155, and Refs 41,42. I also added discussions about not all oligomers are equally neurotoxic in line 441-450 (added Refs 148-151).

> I would appreciate if the authors could include or discuss the effect of metal ions on the lipid-Abeta interaction which might be also important for aggregation or the progression of the disease (e.g. Journal of Biological Chemistry 278(5):2977-82)

     Thank you very much for your suggestion. I agree that lipid-protein and lipid-metal interactions are important. I think ganglioside-containing membrane raft may be the proper platform when amyloid and metals can interact, and that metals (with positive charges) can be concentrated in gangliosides (with negative charges). I added the comments in Introduction (line 56-61) and discussed in line451-455 (added Refs. 153-156). Further, we modified figure3 to exhibit the oligomerization occurs in the membrane surfaces.

Reviewer 3 Report

GENERAL COMMENTS

This review concerns disease-related proteins of different neurodegenerative disorders, focusing mainly on their regulatory role of metal homeostasis as well as on the neurotoxicity triggered by amyloidogenic protein oligomers. The authors analyse the literature and propose hypothesis/schemes for the neurodegeneration process and synaptotoxicity/neurotoxicity (Figures 1 and 3), following a previous review published by them 3 years ago (ref. 3).

Overall, the review seems to be quite complete and the authors have also experience in this field, but some queries must be addressed, as mentioned in Detailed Comments, as well as English revision. In my opinion, the chosen topic meets the interests of the readers of Molecules and the MS deserves to be published with minor corrections.    

DETAILED COMMENTS

  • Please specify, in the Introduction part, the period of revision of the literature (last 2 decades?) and explain the interest of proposing this MS when the same authors made a review under quite similar theme 3 years ago (ref. 3).
  • In 2.5, the second paragraph (from Line 222) must be rewritten in a clear way for understanding of the readers and also because some repetitions from 2.1 are present, namely concerning ZnT-1 transporter.
  • In Figure 1, ZIP-10 is present but the explanation mentions that ZIP-14 transports Fe2+. So, what about ZIP-10?
  • In Figure 3, remove the coloured circle corresponding to Fe because it does not appear in that figure. Correct “Neutotoxicity” to “Neurotoxicity”.
  • In Page 11 (Line 388) explain why Ca2+ channels trigger ROS formation.
  • In Conclusions (Line 421) substitute “some reagents” by “some compounds”.

Author Response

Thank you very much for your comments and suggestions. Considering your comments, I re-organized and polished up my manuscript, and I believe, it can be improved so much.

> Please specify, in the Introduction part, the period of revision of the literature (last 2 decades?) and explain the interest of proposing this MS when the same authors made a review under quite similar theme 3 years ago (ref. 3).

     I apologize for lack of clarity in my manuscript. In the previous review, we had focused the loss of normal protective functions as common neurodegenerative mechanism. In this manuscript, I updated the loss of function hypothesis based on recent literatures, and combined the gain of toxic function hypothesis based on the common features of Ca-permeable pores. I re-organized the most part of Introduction (line 41-45, line 50-64). I also added the newer explanations about lipid-metal interactions and about metal contribution in the hypothesis (line439-455). To avoid the similarity, we deleted Table1. Moreover, we added references 2,3 and other 25 references.

> In 2.5, the second paragraph (from Line 222) must be rewritten in a clear way for understanding of the readers and also because some repetitions from 2.1 are present, namely concerning ZnT-1 transporter.

     I am sorry for the incomplete writing. I made re-organize the part (line 245-254) more clearly, and asked to correct my english by Edanz (www.edanzediting.com/ac, shown in Acknowledgement).

>In Figure 1, ZIP-10 is present but the explanation mentions that ZIP-14 transports Fe2+. So, what about ZIP-10?

     I am sorry for the mistake. It was replaced with DMT1/ZIP14, based on the Ref 71. Ferrireductase can convert Fe3+ to Fe2+, and then, Fe2+ is transported intracellularly through the complex of DMT1/ZIO14. We also added comments in line207 and in figure legend (line 280).

> In Figure 3, remove the coloured circle corresponding to Fe because it does not appear in that figure. Correct “Neutotoxicity” to “Neurotoxicity”.

     Thank you for your correction. I corrected as suggested. According to other reviewer’s comments, Mn was added in Fig.3.

> In Page 11 (Line 388) explain why Ca2+ channels trigger ROS formation.

     I am sorry for the lack of clarity. The membrane disruption will occur after Ca2+ increase, and the resulting lipid peroxidation will produce ROS. I added comments in line302-303.

> In Conclusions (Line 421) substitute “some reagents” by “some compounds”.

     Thank you for your comments. I corrected as suggested.

Round 2

Reviewer 2 Report

The authors have addressed all my comments and concerns. In my opinion the manuscript is now suitable for publication. 

Author Response

Thank you very much.

Owing to your suggestions, I could make my review more proper.

Best wishes